# Ploidy and recombination proficiency shape the evolutionary adaptation to constitutive DNA replication stress

**Marco Fumasoni** [1,2]*, **Andrew W. Murray** [1]*

**1** Department of Molecular and Cellular Biology, Harvard University, Cambridge, Massachusetts, United States of America, **2** Instituto Gulbenkian de Ciência, Oeiras, Portugal

* mfumasoni@igc.gulbenkian.pt (MF); awm@mcb.harvard.edu (AWM)

## Abstract

In haploid budding yeast, evolutionary adaptation to constitutive DNA replication stress alters three genome maintenance modules: DNA replication, the DNA damage checkpoint, and sister chromatid cohesion. We asked how these trajectories depend on genomic features by comparing the adaptation in three strains: haploids, diploids, and recombination deficient haploids. In all three, adaptation happens within 1000 generations at rates that are correlated with the initial fitness defect of the ancestors. Mutations in individual genes are selected at different frequencies in populations with different genomic features, but the benefits these mutations confer are similar in the three strains, and combinations of these mutations reproduce the fitness gains of evolved populations. Despite the differences in the selected mutations, adaptation targets the same three functional modules in strains with different genomic features, revealing a common evolutionary response to constitutive DNA replication stress.

**Data Availability Statement:** All relevant data are within the manuscript and its Supporting Information files. A major dataset, containing the sequencing data used in the manuscript has been made publicly available at the EBI European

## Author summary

Predicting the outcome of an evolutionary process requires precise knowledge of all the variables at play during adaptation. This goal is challenging to achieve since we often lack complete knowledge on which aspects of the organisms selected will influence their evolutionary response. Over the last decade, experiments in budding yeast have shown that the evolutionary adaptation induced by mutations that perturb important cellular processes can follow reproducible trajectories. We asked how genomic features influenced these trajectories by subjecting budding yeast strains with different features, such as ploidy and recombination proficiency, to the same mutation that imposes constitutive DNA replication stress. While we found many differences among the genes selected in different strains, we uncovered a general, feature-independent evolutionary trajectory which alters the same three cellular processes: DNA replication, the DNA damage checkpoint, and the linkage between sister chromosomes. Our results suggest that differences in genomic features control which genes adaptation modifies but have less effect on which biological processes are modified in response to genetic perturbations. We propose that this

Nucleotide Archive (Accession no: PRJEB42058). The custom pipeline used for the data analysis present in figures and supplementary tables are available on GitHub: https://github.com/marcofumasoni/Fumasoni_and_Murrray_2020.

**Funding:** MF was supported by fellowships from The European Horizon 2020 Framework Programme, H2020-MSCA-IF-2020 - 101030203/GENMAINEVO (https://ec.europa.eu/), the Human Frontier Science Program - LT000786/2016-L (https://www.hfsp.org/), the European Molecular Biology Organization - ALTF 485-2015 (https://www.embo.org/) and the Associazione Italiana per la Ricerca sul Cancro - iCARE 17957 (https://www.airc.it/). AWM was supported by grants from the National Institute of General Medical Sciences - GM43987/RO1 (https://www.nigms.nih.gov/), the National Science Foundation - #1764269 (https://www.nsf.gov/) and the Simons Foundation - #594596 (https://www.simonsfoundation.org/). The funders had no role in study design, data collection and analysis, decision to publish, or preparation of the manuscript.

**Competing interests:** The authors have declared that no competing interests exist.

knowledge will increase our ability to predict similar adaptive processes outside the laboratory, including cancer progression, which is characterized by the somatic evolution of cells experiencing DNA replication stress.

## Introduction

Organisms and cells adapt to selective pressures through the acquisition of beneficial mutations. The interactions between beneficial mutations that produce evolved phenotypes are complex and difficult to determine both in laboratory and natural evolution. These interactions are also likely to depend on two factors. The first, genetic background, refers to the collection of individual sequence variants in a strain or population [1–4]. The second is the functional features of the genome including its ploidy and rate of recombination. Previous work has revealed that different genetic backgrounds can select different spectra of mutations and different ploidies lead to different rates of adaptation [5,6,7–14,15,16] but we lack a general, mechanistic explanation of these differences. Here, we take advantage of our earlier work on adaptation to DNA replication stress to investigate how different genomic features control adaptive evolutionary trajectories.

We previously characterized the evolutionary adaptation of haploid cells of the budding yeast, *Saccharomyces cerevisiae*, experiencing a form of constitutive DNA replication stress [17], a perturbation of DNA synthesis that produces lesions in DNA, increases genome instability, and decreases fitness [18–21]. DNA replication stress is often present in cancer cells as a consequence of oncogene activation [22–25] and can also be generated by the unrestrained proliferation of selfish genetic elements such as viruses or transposons [26–29]. In haploid *S. cerevisiae*, evolutionary adaptation to constitutive DNA replication stress is driven by mutations affecting three different functional modules that contribute to genome maintenance: gain-of-function mutations that stabilize the linkage between sister chromatids, mutations that improve the completion of DNA replication, most likely by stabilizing the progression of replication forks, and mutations that inactivate the DNA damage checkpoint, thus accelerating the cell division cycle. Full adaptation to the defects imposed by constitutive DNA replication stress requires a combination of mutations in all three modules, which are acquired in an order dictated by the epistatic interactions between the mutations [17].

We asked how genomic features affected evolutionary adaptation to constitutive DNA replication stress by testing i) diploid cells, which carry two copies of every chromosome and thus make fully recessive mutations phenotypically silent, ii) haploid, recombination-deficient cells which cannot produce gene amplifications or deletions that depend on homologous recombination between repeated sequences [30,31]. These features are relevant for cancer biology: cancer progression begins in diploid cells and defects in homologous recombination have been implicated in predisposition to tumorigenesis (i.e. BRCA2 mutations, [32,33]) and exploited for anticancer therapies [34–36].

To investigate the influence of genomic features, we evolved diploid and recombination-deficient haploid strains and compared their responses with those of haploid, recombination-proficient cells. All three strains experienced constitutive DNA replication stress caused by removing Ctf4 [17], a protein that physically links different activities at replication forks [37]. Adaptation to DNA replication stress occurs in all three strains to extents that are correlated to their initial fitness defects. Whole genome sequencing of the evolved populations revealed diverse mutational spectra with only modest convergence among the three different strains. Despite the differences in which genes were mutated and differences in their genomic features,

all three strains accumulated mutations in the DNA replication machinery, the DNA damage checkpoint, and sister chromatid cohesion. Individually engineering a subset of these mutations into all three strains demonstrated their ability to reduce the fitness cost of DNA replication stress, even in those where they were not selected. For each combination of genomic features, combining mutations in all three modules produced fitness increases that matched those of the corresponding evolved strains. We conclude that although differences in the observed frequency of mutations in different genes depend on genomic features, the principles that govern adaptation to constitutive DNA replications stress are largely feature-independent. We discuss the implication of our results for tumor evolution and for generating differences in the mechanisms of genome maintenance between species.

## Results

### All populations adapt to constitutive DNA replication stress

DNA replication stress is often induced with drugs or by reducing the level of DNA polymerases [38–41]. To avoid evolving drug resistance or increased polymerase expression, which would rapidly overcome DNA replication stress, we deleted the *CTF4* gene, which encodes a non-essential subunit of the DNA replication machinery (the replisome) [42]. Ctf4 is a homotrimer that functions as a structural hub within the replisome [37,43] by binding to the replicative DNA helicase, primase (the enzyme that makes the RNA primers that initiate DNA replication), and other accessory factors [37,44–46]. In the absence of Ctf4, the Polα-primase and other lagging strand processing factors are poorly recruited to the replisome [37,44,47], causing several characteristic features of DNA replication stress, such as accumulation of single strand DNA (ssDNA) gaps [48,49], reversed and stalled forks [17,49], cell cycle checkpoint activation [47,50] and altered chromosome metabolism [51,52]. The constant presence of these defects in *ctf4Δ* cells produces a constitutive form of DNA replication stress, which results in a substantially reduced reproductive fitness [17]. We removed *CTF4* from wild-type (WT) haploid and diploid cells, as well as from haploids impaired in homologous recombination due to the deletion of *RAD52* (Fig 1A), which encodes a conserved enzyme required for pairing homologous DNA sequences during recombination [53]. Because Rad52 is involved in different forms of homologous recombination, it's absence produces the most severe recombination defects and thus allows us to achieve the largest recombination defect achievable with a single gene deletion [54]. Attempts to generate homozygous *ctf4Δ/ctf4Δ rad52Δ/rad52Δ* mutants failed due to the lethality of the resulting zygotes. We refer to the strains we analyzed as haploids (*ctf4Δ*, yellow), diploids (*ctf4Δ/ctf4Δ*, blue) and recombination-deficient (*ctf4Δ rad52Δ*, red). Compared to the previously reported *ctf4Δ* haploid strains, which manifested a fitness decrease of ~27% relative to isogenic wild-type cells [17], constitutive DNA replication stress caused by the absence of Ctf4 was more detrimental in diploid (~35% reduced fitness) and recombination-deficient strains (~55% reduced fitness) (Fig 1B). Diploid cells require twice as many forks as haploids and Ctf4-deficient diploids are thus more likely to have forks that cause severe cell-cycle delays or cell lethality. We speculate that this increased probability explains the more prominent fitness defect displayed by diploid cells. Interestingly, homologs of Ctf4 are absent in prokaryotes, where the primase is physically linked to the replicative helicase [55] and Ctf4 is essential in the cells of eukaryotes with larger genomes such as chickens [48] and humans [56]. Rad52 is likely involved in rescuing stalled replication forks by recombination-dependent mechanisms [49,57]. We speculate that the absence of Rad52 increases the duration of these stalls and leads some of them to become double-stranded breaks resulting in cell lethality and explaining the decreased fitness of *ctf4Δ rad52Δ* haploid double mutants. In diploids *ctf4Δ rad52Δ* cells, which have twice as many chromosomes, the number of

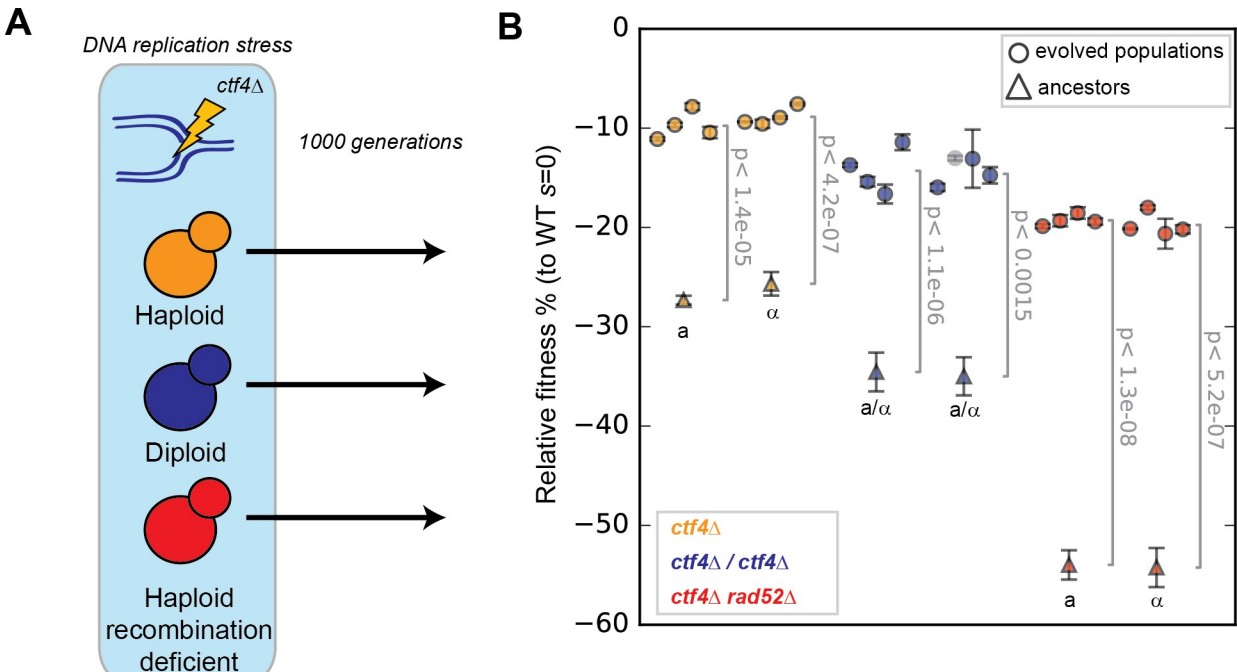

**Fig 1. Evolutionary adaptation to constitutive DNA replication stress in strains with different genomic features. (A)** Experimental scheme: Constitutive DNA replication stress was induced by removing Ctf4, a protein that coordinates replisome activities. Haploid (orange), diploids (blue) and cells with a severely decreased ability to perform homologous recombination (red, recombination-deficient), were subjected to 1000 generations of experimental evolution in the presence of constitutive DNA replication stress. **(B)** The fitness of the 6 ancestral strains (2 for each genome architecture) and of 24 evolved populations derived from them, relative to WT cells (s = 0). Cells were competed against reference strains with the same ploidy. Fitness data of haploid strains (orange) is from [17]. The semi-transparent gray dot represents a diploid strain that became haploid during the experiment and is thus excluded from subsequent analysis. Error bars represent standard deviations. **a** and α refer to the strains' mating type (*MAT* locus). a/α indicates diploid strains. The P-values reported in figures are the result of t-tests assuming unequal variances (Welch's test). The fitness values shown here are reported in S1 Data.

irreparably stalled fork may be sufficient to kill most of the cells in a population, thus explaining the unviability of the strain.

We evolved eight parallel diploid and recombination-deficient populations for 1000 generations in the same conditions and at the same time as the haploid populations we previously analyzed [17] (Fig 1A and Materials and methods). In this regime, spontaneous mutations that increase fitness and survive genetic drift will spread through the population [58,59]. In addition, we evolved eight wild-type diploid populations like the previously evolved wild-type haploids, to control for mutations that are selected by the culture conditions rather than by ameliorating DNA replication stress [17].

After 1000 generations experiencing constitutive DNA replication stress, we asked how the fitness of the evolved diploid and recombination-deficient populations compared to those of evolved haploids. Regardless of their genomic features, all populations lacking Ctf4 increased in fitness compared to their respective ancestors: haploids from -27% to -9.6 ± 1.2% [17], diploids from -35% to -14.4 ± 1.8%, and recombination-deficient haploids from -55% to -19 ± 1% (Fig 1B). The fitness of four clones isolated from each evolved population were similar to the fitness of the population they were derived from (S1 Fig). One diploid line (EVO14) gave rise to a population with a haploid genome content, suggesting a possible haploidization event during evolution. Sequencing revealed no aneuploidies as a potential explanation of this phenomenon. While diploidization has been recurrently observed during experimental evolution with budding yeast [16,60–62], reports of spontaneous haploidization events have been instead

scarce. Given the difficulties introduced by the change of ploidy over the 1000 generations, we have excluded EVO14 from all our analyses. Control evolved wild-type populations showed only small fitness increases: 4 ± 1% for haploids and 6.7 ± 2.7% for diploids (S2 Fig).

## The rate of adaptation to DNA replication stress depends on the ancestor's fitness

The dynamics of adaptation of strains with different genomic features was followed by measuring the fitness of samples of all populations saved during the course of the evolution experiment (Fig 2A). Although all populations increased their fitness in response to constitutive DNA replication stress, strains with different genomic features increased in fitness to different extents (measured as the absolute fitness gain of populations over 1000 generations). Recombination-deficient populations showed the largest fitness gain (+34.6 ± 0.8%), followed by diploid (+20.3 ± 1.6%) and haploid populations (+17.2 ± 1.0%, Fig 2B). Interestingly, we found that these values mostly depend on different adaptation rates during the first 100 generations. Diploids and recombination deficient populations gain a large fraction of their final fitness during this period, approaching the initial fitness of the evolving haploid populations. Afterwards, all the populations adapted at comparable rates for the reminder of the experiment (Fig 2C). We asked what could have generated different adaptation rates in response to the same selective pressures. Less fit ancestral strains have been proposed to increase in fitness faster than more fit ancestors, a phenomenon referred as 'declining adaptability' [63–66]. To test for this possibility, we plotted the final fitness of all evolved populations against the fitness of their respective ancestors (Fig 2D). We found that the absolute fitness gain over 1000 generations depended linearly on the ancestors' relative fitness ($R^2$ = 0.95). These results agree with previous studies, which observed declining adaptability across a wide range of organisms and selection conditions [9,10,67,68].

The majority of previous studies have suggested that diploids adapt more slowly than isogenic haploids [13,69,70]. Our diploid *ctf4Δ/ctf4Δ* populations gained more fitness than *ctf4Δ* haploids over the course of 1000 generations (Fig 2B).

Wild-type diploids increased more in absolute fitness than wild-type haploids (Fig 2E), suggesting that the discrepancy between our work and previous studies is not due to the presence of replication stress and the associated increase in mutation rate. Diploid populations recovered a slightly smaller fraction of their initial fitness defect compared to haploid and recombination-deficient strains (S4A Fig). Collectively, our results suggest that ploidy is a poor predictor of the speed and extent of adaptation (S3 Fig). We conclude that the fitness of the ancestral cells is the major determinant influencing how much populations adapt, adding further support to the concept of declining adaptability.

## Limited parallelism in adaptively mutated genes across genomic features

We used whole-genome sequencing to understand the genetic basis of the evolutionary adaptation to DNA replication stress. For each of the three strain with differing genomic features, we sequenced all evolved populations, as well as four individual clones isolated from each of the *ctf4Δ* populations. Genes whose mutations were significantly selected in the evolved wild-type populations were removed from the list of evolved *ctf4Δ* strains to control for mutations selected by the environment of the experiment, rather than the absence of Ctf4 (S6 Fig, see material and methods for details). After applying this filter, a total of 195, 276 and 905 genes or associated regulatory sequences were found mutated in evolved haploid, diploid, and recombination-deficient populations respectively (Fig 3A and S2 Table). When normalized for DNA content, haploid (13 mutations/haploid genome) and diploid clones (9 mutations/haploid

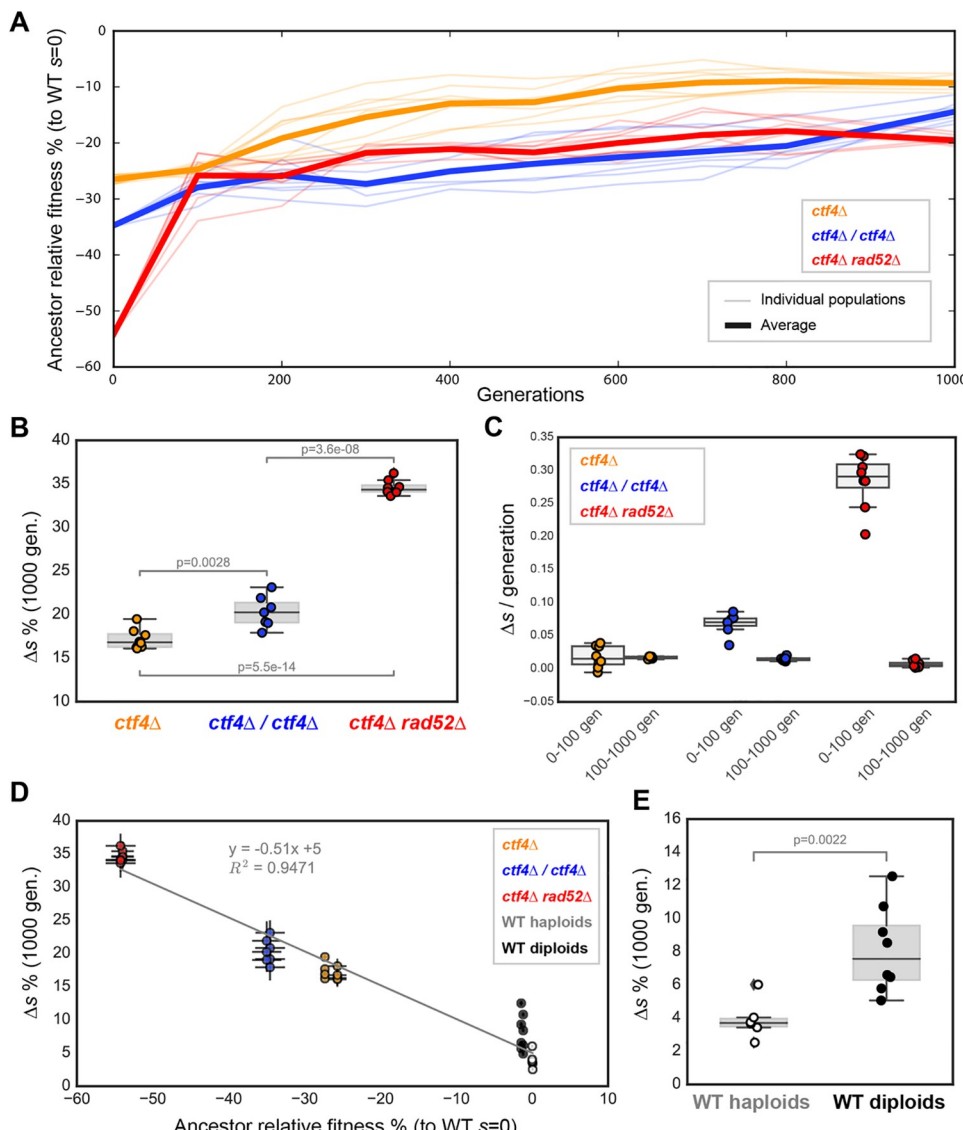

**Fig 2. Fitness increase over 1000 generations. (A)** Fitness change over the course of evolution for strains with *ctf4Δ*-induced replication stress. Average trajectories for populations with different genomic features are shown as thick lines. Thin traces represent individual populations' trajectories. **(B)** Absolute fitness increase over 1000 generations for the evolved populations. Δ*s* is the difference between the fitness of the evolved populations and the fitness of their ancestors, with both fitnesses measured relative to that of cells that contain Ctf4 (*CTF4*) and have the same ploidy. **(C).** Adaptation rates (measured as Δ*s*/generation) during the early phase of evolution (0 to 100 generations) and the rest of the experiment (100 to 1000 generations). **(D)** Shows the correlation between the fitness increase during evolution (Δ*s*) and the fitness defect of the ancestors relative to cells of the same architecture containing Ctf4 (the ancestor relative fitness). **(E)** Absolute fitness increase over 1000 generations for control strains that contained wild-type *CTF4*. Fitness data of haploid strains (orange and white) is from [17]. Error bars represent standard deviations. Grey line represents linear regression between the datapoints. The P-values reported in figures are the result of t-tests assuming unequal variances (Welch's test). The data shown here are reported in or derived from S1 Data.

genome) had similar but statistically different mutation frequencies (p = 4x10⁻⁶, Welch's test, S5A Fig) but recombination-deficient clones carried many more mutations, both in total (50/ haploid genome, S5A Fig) and as synonymous mutations (7/haploid genome, S5B Fig). Our results agree with the reported increase in spontaneous mutagenesis in *rad52Δ* cells [71–73].

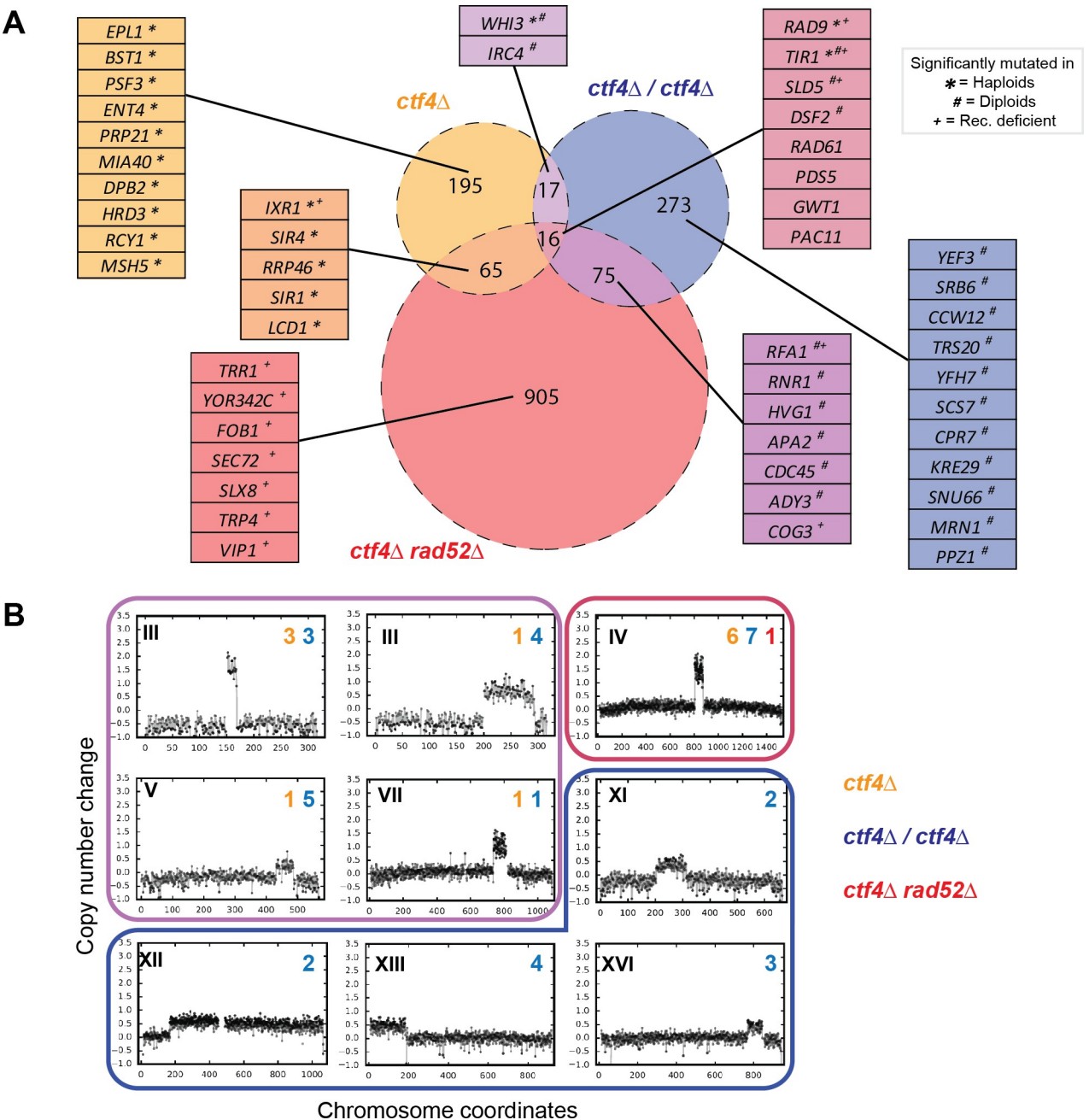

**Fig 3. Putative adaptive mutations in the evolved clones.** (A) Venn diagram representing the mutations (SNPs and small InDels 1-55bp) found in the evolved clones and populations. The area of the circles is proportional to the list of genes mutated at least once in the set of evolved clones for each genomic feature. The number within each sector represents number of genes which were mutated in one, two, or three genomic features. Tables include the genes whose mutations were significantly mutated in either global or feature-specific analysis. Symbols next to the genes highlight that they were statistically significantly mutated in strains with individual features (* refers to haploid, # to diploid and + to recombination-deficient features, e.g. *SLD5* <sup>+#</sup> means that the gene is found mutated in strains with all three features, but is statistically significantly mutated only in diploid and recombination-deficient populations). The identities of all the genes in the Venn diagram intersections are reported in S5 Data (B) Copy number variations (CNVs) affecting different chromosomes (roman numbers) which appeared in at least two independent populations. Arabic numbers are the number of independent populations, in each genomic feature (highlighted by a different color), in which the CNVs were detected. Copy number change refers to the fragment's gain or loss during the evolution experiment (i.e. +1 means that one copy was gained in haploid cells and that two copies were gained in diploid cells). Frames around the plots have the same color as the sectors in the Venn diagram, and encircle CNVs that have been found in one, two, or three genomic features. All the segmental amplifications detected are reported in S4 Table.

We used statistical analysis to identify candidate adaptive mutations, by identifying genes that were mutated more often than expected by chance in independent populations (see Materials and methods for details). We performed this analysis both on individual strains and the collection of three strains with different genomic features. Using a false discovery rate of 5% we found a total 50 genes that met one or both criteria. Interestingly, we found no significant overlap (p = 0.48, hypergeometric test) between these genes and those previously reported to have positive genetic interactions with ctf4Δ mutants [74], highlighting how systematic analysis of genetic interactions fails to predict the outcome of an evolutionary repair process.

Only 8 of the putatively selected genes were mutated in all three strains (Fig 3A). This small number could be due to strain-specific adaptive strategies adopted, differences in the rate for the same mutation in different strains, differences in the selective benefit the mutation conferred, or a combination of these three effects. We found 28 genes with putative adaptive mutations that are specific to individual strains, as well as 14 genes with putative adaptive mutations in two strains (Fig 3A, genes highlighted in boxes and S2 Table). Analyzing gene ontology (GO) terms associated with the putative adaptive mutations revealed an enrichment of biological process terms associated with genome maintenance (S3 Table). The genes that were significantly mutated in all three strains included two that we had previously demonstrated to be adaptive in haploid ctf4Δ cells: mutations in *RAD9* inactivate the DNA damage checkpoint and those in *SLD5* alter the function of the replicative helicase [17]. All three strains also had mutations in *RAD61* and *PDS5*, which are implicated in chromosome segregation [75,76], a cellular module shown to play a role during the adaptation of haploid ctf4Δ populations [17].

We also examined the evolved genomes for changes in gene copy numbers (Copy Number Variations, CNVs) that might have played a role in adaptation. Many clones displayed an increased copy number of defined chromosome segments (segmental amplification) compared to the control populations (S4 Table). We failed to detect an enrichment of any biological process GO-terms, or of reported dosage-sensitive genes on these amplified segments ([77], S5 Table). Segmental amplifications were common in diploid clones (S5C Fig). We inferred that segmental amplifications were putatively adaptive if the same chromosome region was found amplified in two clones isolated from independent populations (Fig 3B). Among these segmental amplifications, four CNVs were found only in evolved diploids, four were found in at least an evolved haploid and a diploid clone, and only one was detected in populations of all three strains we evolved (Fig 3B). This latter fragment is located on chromosome IV and contains *SCC2*, which encodes a subunit of the cohesin loader complex. This essential complex is responsible for loading the proteinaceous cohesin ring that holds sister chromatids together until they segregate from each other during mitosis [78–80]. The gene encoding for the other subunit of the cohesin loader complex, *SCC4*, lies on the segment of chromosome V that was amplified in both haploid and diploid clones. *SCC4* was never found amplified alone, but always in populations that had also amplified *SCC2* (S7A Fig). We previously showed that extra copies of *SCC2* alleviate the cohesion defects associated with the absence of Ctf4 in haploid cells and increase their fitness; the additional amplification of *SCC4* further increase cells' fitness, but there is no benefit when *SCC4* is amplified alone [17]. These observations are compatible with a scenario where Scc2 is the limiting subunit in the formation of active cohesin loader complexes (S7B Fig).

Overall, genome sequencing reveals that the identity of the genes carrying selected mutations varies depending on cells' genomic features. Nevertheless, we found evidence of common evolutionary strategies across different populations: Mutations affecting sister chromatid cohesion, DNA replication, and the DNA damage checkpoint, which were previously shown to drive adaptation to a form of DNA replication stress in haploid cells [17], are significantly mutated regardless of the genomic features of the ancestral cells.

## Evolved diploids show adaptive loss of heterozygosity

We analyzed the frequency of mutated reads in evolved diploid clones to estimate whether mutations were homozygous or heterozygous (S8 Fig). We detected 36 homozygous mutations across the evolved diploid clones (Figs 4A and S8B and S6 Table). Homozygous mutations were evenly distributed across the evolved clones analyzed and thus were not the product of a single unrepresentative event (S8C Fig). Individual clones often carried multiple homozygous mutations on the same chromosome arm (S6 Table) suggesting that the loss-of-heterozygosity (LOH) was likely the consequence of mitotic recombination events affecting large chromosomal segments [81]. In addition to homozygous mutations, three genes were found to carry two different mutations, most likely one in each of the two copies of the gene (double hits, Fig 4A and S6 Table). Ten of the homozygous or double hit mutations were abundant within their respective populations (mutation penetrance > 70%, Figs 4B and S9). Gene ontology analysis of the 39 genes with homozygous mutations or double hit alleles revealed the enrichment of many terms associated with genome maintenance (S7 Table). One cluster of genes, linked by genetic and physical interactions, affected components of the replisome and factors involved in the DNA damage checkpoint (Fig 4B). Because these genes had been changed by two events (either two mutations, or a mutation followed by LOH) and the altered forms had risen to high frequency in the evolved populations (Fig 4B), we argue that the inactivation or modification of many of these genes was selected for by constitutive DNA replication stress.

## Genomic features influence the distribution of adaptive mutations

In addition to altering which genes get adaptive mutations, we hypothesize that genomic features could affect evolutionary trajectories in two ways: changing the frequency at which different modules are altered and changing the distribution of the types of mutations that are selected (for example by altering the ratio of adaptive point mutations to copy number variation [73,82–86]). Our analysis revealed that all three strains, which differed in their genomic features, acquired putative adaptive mutations in three genome maintenance modules: sister chromatid cohesion, DNA replication and the DNA damage checkpoint. Out of the 23 populations analyzed, 18 of them contained at least one clone with mutations affecting all three of these modules, while all of them contain a clone with mutations in two of the three modules (Fig 5A). Despite the ubiquity of mutations in genome maintenance, the frequency of mutations in genes known to affect individual cellular modules varied between the strains: mutations affecting sister chromatid cohesion were underrepresented in recombination-deficient strains (p = 2.7e-04, $\chi^2$ test) and there were fewer mutations affecting the DNA damage checkpoint in diploid clones (p = 0.03, $\chi^2$ test, Fig 6A).

If amplifying the genes for the cohesin loader, *SCC2* and *SCC4*, depended on homologous recombination between repeated sequences, the rate of amplification should be dramatically decreased in the recombination-deficient strain, which lacks Rad52. Analysis of the segmental amplifications containing *SCC2* or *SCC4* revealed that their boundaries correspond to three classes of repeated sequences: transposons (Ty elements), transposon-associated long terminal repeats (LTRs) and tRNA genes (S10 Fig and S8 Table). Recombination at these repeated elements has been previously shown to produce the segmental amplifications that are selected by limited nutrient availability [87–89]. Amplification of *SCC2* thus occurs as a result of homologous recombination between two different copies of the repeated sequences on chromosome IV, one on either side of *SCC2* and amplification of *SCC4* is due to similar events on chromosome V. This dependence on homologous recombination explains the absence of segmental amplifications in all but one of the recombination-deficient clones (Fig 3C and S4 Table); this single event did not occur by recombination between different copies of a repeated sequence.

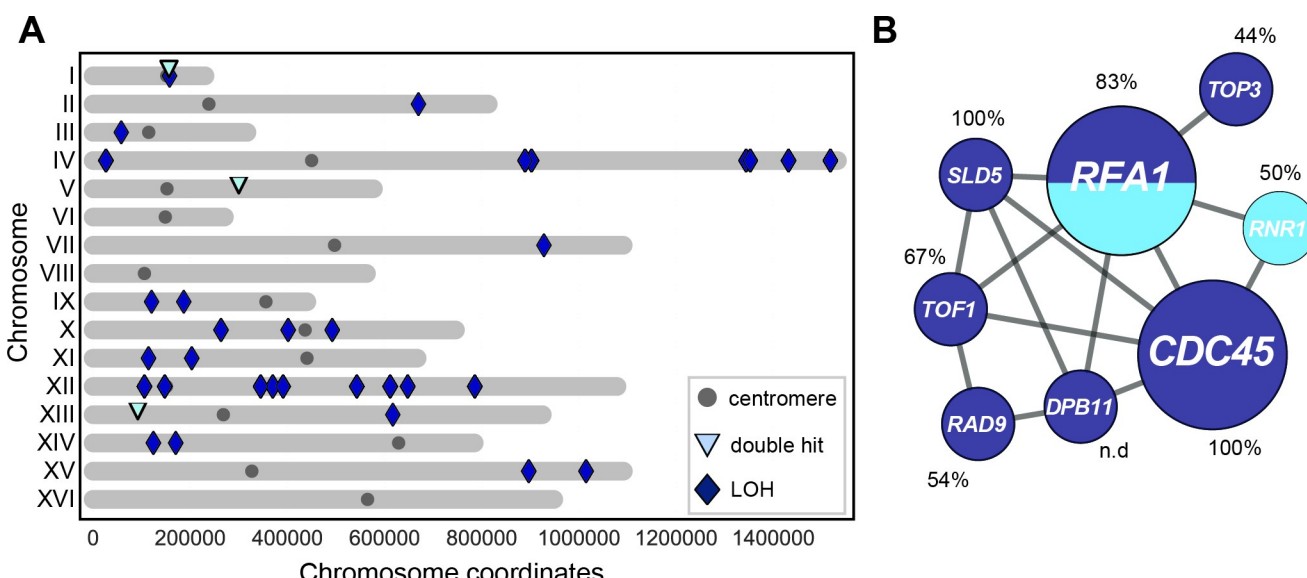

**Fig 4. Homozygous mutations in diploids evolved in the presence of constitutive DNA replication stress. (A)** Homozygous mutations (SNPs and small InDels of 1-55bp) detected in evolved *ctf4Δ/ctf4Δ* diploids clones distributed by chromosome (roman numbers). Dark blue diamonds represent loss of heterozygosity events (LOH). Light-blue triangles represent genes which contain two independent mutations (double hits) and the gray circles are the centromeres. The identities of the mutations shown here are reported in S6 Table. **(B)** DNA replication and DNA damage checkpoint genes with mutations in both copies in evolved *ctf4Δ/ctf4Δ* diploid clones. Gray lines represent evidence of genetic and physical interactions from the literature (https://string-db.org). Node diameter is proportional to the number of populations in which the gene was mutated. Dark blue represents LOH events, light blue represents double hits. Percentages refer to the average estimated frequency within populations of the clones carrying the respective mutation in homozygosity (LOH) or carrying both mutations (double hits).

Because changes affecting sister chromatid cohesion rely predominantly on gene amplifications (Fig 5C), we detect fewer mutations affecting this module in recombination-deficient strains (Fig 5A).

In haploid *ctf4Δ* cells, inactivating the DNA damage checkpoint increases fitness [17]. The primary target for mutation is *RAD9*, both because of its large size (3927 bp) and the presence of a run of 11 As, which repeatedly gain or lose an extra nucleotide causing a frameshift mutation. Because this is a loss-of-function mutation [17], removing Rad9 function in diploids requires two events, thus reducing the probability of adapting by destroying the DNA damage checkpoint. Despite this difficulty, one diploid clone is homozygous for the frameshift mutation in *RAD9*, most likely as a result of a mutation in one copy of the gene followed by mitotic recombination or gene conversion that homozygosed this mutation and fully inactivated the checkpoint. The need to follow a two-hit pathway to fully inactivate the DNA damage checkpoint explains why fewer *RAD9* mutations were found in diploids.

We conclude that although mutations affecting sister chromatid cohesion, DNA replication and DNA damage checkpoint are found in all three strains, the frequency at which we detect them depends on the genomic features of an evolving population: gene amplification occurs much less frequently in recombination-deficient populations, making one pathway for improving sister chromatid cohesion less accessible, and recessive null mutations are rarer in diploid populations, blocking a major route by which haploids inactivate the DNA damage checkpoint.

## Mutations affecting sister chromatid cohesion, DNA replication and DNA damage checkpoint recapitulate evolution in all genomic features

We asked whether genomic features affected the benefit that mutations conferred. To test this hypothesis, we chose a group of recurring mutations to validate their causality and measure

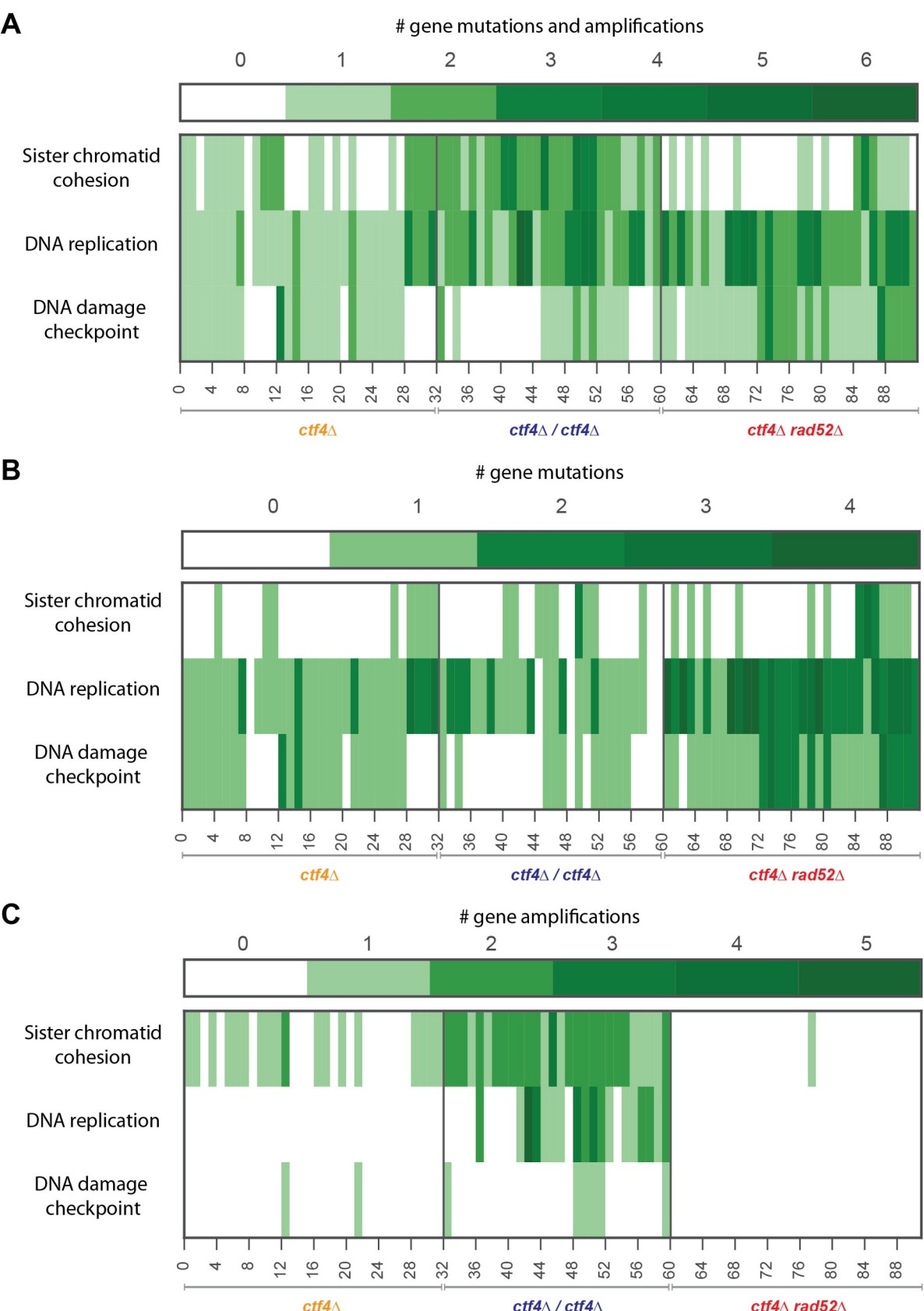

**Fig 5. Mutations affecting sister chromatid cohesion, DNA replication and the DNA damage checkpoint. (A)** Heatmaps representing all the mutations (SNPs, small InDels of 1-55bp and segmental amplifications) causally implicated in these phenotypes in the 92 sequenced evolved clones. **(B)** Frequency of SNPs and small InDels (1-55bp) affecting genes (Open reading frames and associated regulatory regions). **(C)** Frequency of genes associated with the three processes that are present on segmental amplifications. The number of hits and the identities of the genes which were mutated are reported in S6 Data.

their fitness benefit in strains with different genomic features. We mimicked the segmental amplification of chromosome IV (Fig 3B), affecting the cohesin loader *SCC2*, by integrating an extra copy of the gene at the *URA3* locus (referred to as *2xSCC2*). Ixr1 is a transcription factor that indirectly affects DNA replication by positively regulating the production of deoxyribonu-cleotide triphosphates (dNTPs, [90]). The abundance of mutations causing premature stop codons in *IXR1* suggested that evolution selected for the inactivation of this gene (S9 Table). We mimicked the loss of a functional Ixr1 protein by gene deletion (*ixr1Δ*). Another recur-rently mutated gene affecting DNA replication was *SLD5*, which encodes the subunit of the main replicative helicase (CMG, [91]) that is bound *in vivo* by Ctf4 [47]. *SLD5* is an essential gene and our earlier analysis suggested that mutations like *sld5-E130K* reduced or altered the helicase's function [17]. Finally, the majority of mutations affecting the DNA damage check-point factor *RAD9* resulted in early stop codons (S10 Table), leading us to test the effect of the *rad9Δ* gene deletion. These mutations were all previously demonstrated to be adaptive in hap-loid cells [17] and were used because they occurred, albeit at different frequencies, in all three strains (point mutations in *SLD5* and *RAD9* and amplification of *SCC2*) or at in at least two (point mutations in *IXR1* in both haploid populations).

We found that all four mutations were adaptive in all three strains. Remarkably, mutations affecting individual genes were adaptive even in strains where they were never (*IXR1* muta-tions in diploids) or rarely detected (*SLD5* mutations in haploids, *RAD9* mutations in diploids and *SCC2* amplification in recombination-deficient strains). Three of the mutations we tested (*SCC2* amplification, *sld5-E130K*, and *rad9Δ*) conferred similar benefits in all three strains, but the fourth, *ixr1Δ*, gave a much larger benefit in the recombination-deficient strain (S11 Fig). One possibility is that the absence of Ixr1 reduces the frequency of double-stranded breaks associated with DNA replication in the absence of Ctf4. These breaks are lethal in the absence of Rad52 but can be repaired in the other two architectures.

For the diploid *ctf4Δ/ctf4Δ* strain, we tested each mutation as a heterozygote (e.g. *IXR1/ixr1Δ*) and a homozygote (e.g. *ixr1Δ/ixr1Δ*). The homozygous deletion of *IXR1* gave a substan-tial fitness benefit (Fig 6A, dark blue), but strains heterozygous for the deletion were slightly less fit than their ancestors (Fig 6A, light blue). For the other three mutations (an extra copy of *SCC2*, *sld5-E130K*, and *rad9*), the heterozygotes conferred a fitness benefit that was significant, albeit weaker than that of the homozygote. These results suggest that homozygous mutations can be selected by a two-step process in which both the first step, mutation in one copy of the gene, and the second, mutation in the other copy or loss of heterozygosity, confer a selective advantage. We conclude that, mutations affecting sister chromatid cohesion, DNA replication and DNA damage checkpoint are beneficial in the presence of constitutive DNA replication stress independently of the genomic features of the strains.

Are mutations in sister chromatid cohesion, DNA replication and the DNA damage check-point enough to recapitulate the evolved fitness increase in all three strains? We previously showed that a haploid *ctf4Δ* strain carrying a mutation in all three of these modules recapitu-lated the fitness of the evolved strains (Fig 6B and [17]). We therefore asked if mutations in these three modules were sufficient to recapitulate 1000 generations of evolution in diploid and recombination-deficient strains. We constructed strains that carried mutations affecting sister chromatid cohesion, DNA replication and DNA damage checkpoint (Fig 6B), using

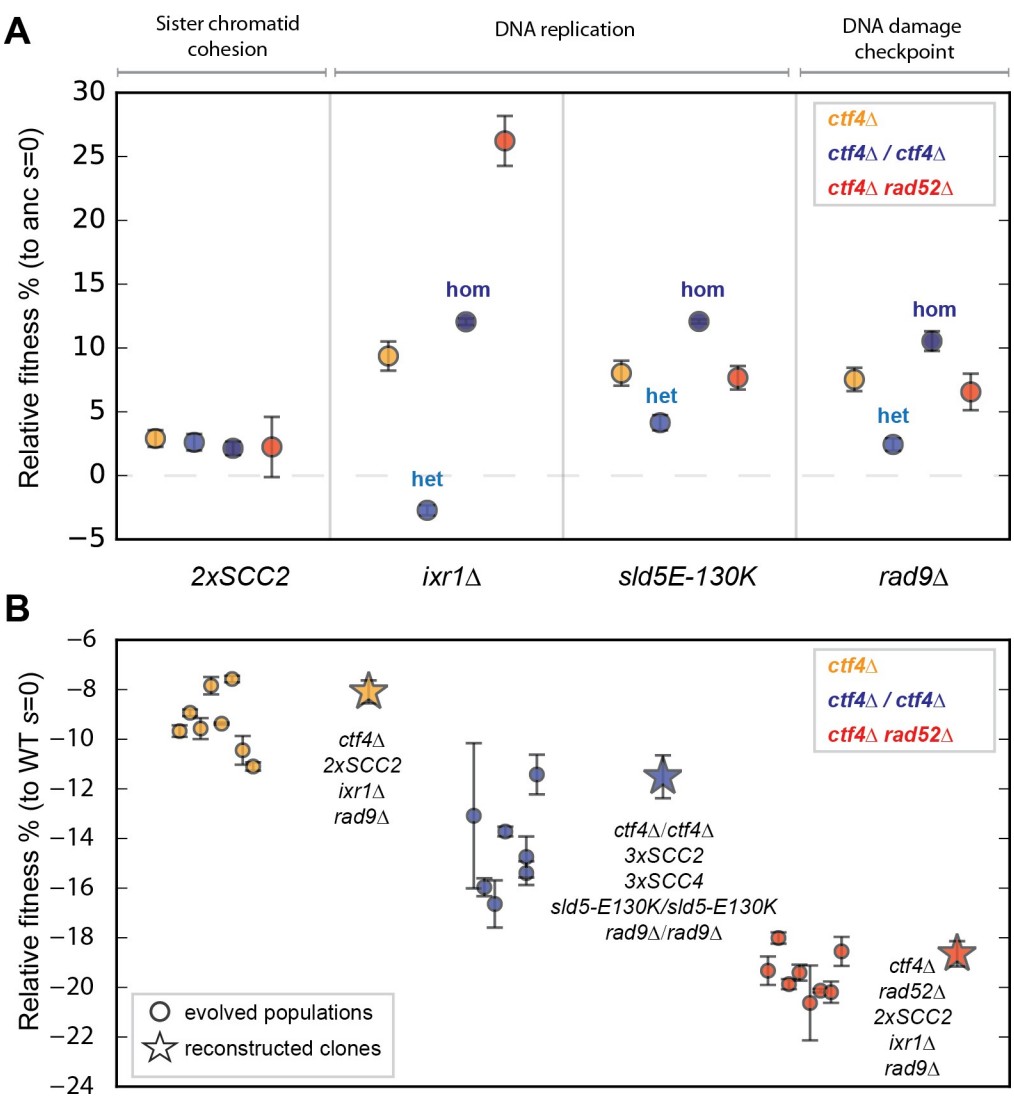

**Fig 6. Reconstructed adaptive mutations can produce evolved fitness increases. (A)** A subset of mutations affecting sister chromatid cohesion, DNA replication and DNA damage checkpoint were re-constructed in the the three ancestral strains that differed in their genomic features. The fitness of the re-constructed strains, relative to their respective *ctf4Δ* ancestor cells (*s* = 0) is depicted. For diploid strains, we replaced one (het) or two (hom) copies of the wild-type gene with the mutant allele. The fitness values shown here are reported in S7 Data. **(B)** Reconstructed strains carrying combinations of adaptive mutations affecting sister chromatid cohesion, DNA replication and DNA damage checkpoint recapitulate the fitness of their respective evolved populations after 1000 generations. The most frequently mutated gene affecting DNA replication was chosen for the reconstruction in individual features (e.g. *IXR1* was mutated more often that *SLD5* in haploids whereas the reverse was true in diploids). Fitness data of haploid strains (orange) is from [17]. The P-values reported in figures are the result of t-tests assuming unequal variances (Welch's test). The fitness values shown here are reported in S7 Data.

mutations that had been found in strains with the same genomic features (Figs 3 and 4 and S1 and S4 Tables) and shown to be adaptive in isolation (Fig 6A).

For all three ancestral *ctf4Δ* strains, their reconstructed derivatives had a fitness that was similar to the evolved descendants of the same strains (Fig 6B). This result demonstrates that for all three combinations of genomic features, acquiring mutations in all three modules (sister chromatid cohesion, DNA replication and DNA damage checkpoint) can recapitulate the

fitness increase seen in response to evolution in the presence of *ctf4Δ*-induced DNA replication stress.

## Discussion

We previously showed that haploid cells adapt to a form of constitutive DNA replication stress by accumulating mutations affecting DNA replication, sister chromatid cohesion, and the DNA damage checkpoint [17]. We repeated this experiment in strains with two other combinations of genomic features, a diploid and a recombination-deficient haploid strain, to ask which aspects of this evolutionary trajectory were general and which were specific to a particular set of genomic features. We find that the genes that acquire adaptive mutations, the frequency at which they are mutated, and the frequency at which these mutations are selected all differ between the three strains with differing genomic features but that mutations that confer strong benefits can occur in all three modules in each strain. Engineering one mutation in each module into an ancestral strain lacking Ctf4 is enough to produce the evolved fitness increase in all three strains. Furthermore, reconstructing a panel of mutations into all three strains proved they are adaptive even in strains where the affected genes were not found significantly mutated by the end of the experiment. Altogether our results demonstrate the existence of a common pathway for yeast cells to adapt to a form of constitutive DNA replication stress.

### The speed of adaptation to constitutive DNA replication stress depends on initial fitness not genomic features

What determines the rate of evolutionary adaptation? One possibility is an organism's genomic features: beneficial, recessive mutations are harder to select in diploids [69,92] and gene amplification is less frequent in recombination-defective cells [93,94], suggesting that recombination proficient haploids should evolve faster than diploids or recombination-defective haploids. Another explanation is that the initial fitness of the population is the principal factor in setting the rate of adaptation, with less fit ancestral populations increasing in fitness faster than more fit ones, a phenomenon referred to as declining adaptability [63]. We observed that the initial adaptation rates were higher for the two less fit strains than they were for the recombination proficient haploids, revealing that that the initial recovery from *ctf4Δ*-induced DNA replication stress follows the declining adaptability documented in other experimental conditions.

Previous work has produced a variety of results with the majority reporting that increasing ploidy decreases the rate of evolution [13,69,70] and a minority reaching the opposite conclusion [12,14]. We suggest that this variability arises because differences in genomic features affects two aspects of the mutations that produce adaptation: the frequency at which they occur and the selective advantage they confer. Thus, we find that defects in recombination reduce the rate of gene amplification and diploidy makes the accumulation of fully recessive mutations require two successive genetic events rather than a single mutation. Despite these barriers, our diploids and recombination-defective haploids initially evolved faster than recombination-proficient haploids. The different mutations rates in strains with different genomic features may affect the adaptation rates of the various populations. Cells lacking Ctf4 were previously reported to have a 5-fold increase in their mutations rate [49], and we found an additional 4-fold increase in recombination-deficient strains (S5A Fig). Nevertheless, it's not clear how much these differences contribute to determining the adaptation rates observed. First, despite having a higher mutations rate, the recombination-deficient strains adapt at a similar pace to haploid and diploid strains for much of the experiment (Generation 100 to 1000, Fig 2A and 2C). Second, faster adaptation in diploids is not limited to populations

experiencing DNA replication stress (Fig 2B) but is also present in control wild-type populations (Fig 2E), which do not have altered mutations rates. Our results thus go against the trend of slower adaptation in diploids as compared to haploids reported by the majority of other studies [13,69,70] and support the idea that the details of genotypes, selections, and experimental protocols can determine the effect of ploidy on adaptation.

We engineered the same mutations in strains with the three combinations of genomic features to ask if the benefit of individual mutations depended on either the genomic features or the fitness of the recipient strain. For three of the four mutations we tested (*sld5-E130K*, *rad9Δ*, and 2x*SCC2*) their effect was independent of both these variables. The fourth (*ixr1Δ*) showed diminishing return epistasis, defined as an adaptive mutation producing a smaller benefit when present in fitter genetic backgrounds (S11 Fig, [65]). This effect has been invoked to explain declining adaptability and is thought to arise from the complex pattern of epistasis between the adaptive mutations and the set of alleles present in the genetic background [68,95–98]. Our results therefore agree with previous reports observing declining adaptability across strains with different initial fitness but largely fail to observe diminishing return epistasis as a potential justification of this phenomenon. Our experiments and two previous evolutionary repair experiments [99,100] both show interactions that are approximately additive between different selected mutations. The reasons for this difference are currently unknown.

## General principles of adaptation to *ctf4Δ*-induced DNA replication stress

Evolutionary change can be examined at multiple levels from the details of adaptive mutations in individual genes, the identity of the functional modules in which the mutations lie, cellular phenotypes, such as the integrity of the DNA damage checkpoint, to overall reproductive fitness. For each of these levels, we examined the effect of genomic features such as ploidy and recombination-proficiency on the adaptation to a form of constitutive DNA replication stress. At both ends of this scale we see substantial variation: genomic features influence the frequency at which mutations occur, the fitness benefit they confer, and the extent of overall adaptation. In the middle we see more conservation, with all three strains, which differed in their genomic features, accumulating mutations in the same three functional modules, suggesting that they all follow similar phenotypic paths by increasing the stability of the linkage between sister chromatids, stabilizing replication forks, and inactivating the DNA damage checkpoint. To test this hypothesis, we reconstructed mutations from all three modules into the three ancestral, Ctf4-deficient strains. This manipulation reproduced fitness observed after 1000 generations of evolution leading us to conclude that the principles that govern evolutionary adaptation to constitutive DNA replication stress are conserved across the combinations of genomic features we examined. Nevertheless, we cannot exclude the possibility that mutations that are only found in the presence of a particular genomic feature may play important roles in adaptation. One example is mutations in the gene encoding the replication fork barrier Fob1, which coordinates the replication of the tandem repeats present at the *rDNA* locus. In the absence of Ctf4, Fob1 can induce replication fork collapse, resulting in double strand breaks which can be rescued by recombination mediated processes. Inactivating Fob1 is thus specifically beneficial in recombination-deficient cells which would otherwise accumulate irreparable chromosome breaks [101].

## Molecular mechanisms affecting adaptation in diploid cells

The phenotype of heterozygous mutations in diploid cells depends on their dominance. Although loss-of-function mutations are often assumed to be fully recessive, a substantial minority have detectable phenotypes as heterozygotes (referred to as semi-dominance or

haploinsufficiency), typically much weaker than that of the homozygous loss-of-function mutations [102]. The phenotype of the heterozygote determines whether mutations will be selected for in diploid populations [103,104]. As an example, loss-of-function mutations in *IXR1* were repeatedly found in haploid and recombination-deficient populations but were not present in evolved diploid strains. This observation is consistent with the slight fitness cost of the heterozygous *IXR1* deletion, despite the strong benefit provided by the homozygous deletion: the cost of the heterozygote acts as a barrier that prevents access to the beneficial homozygous mutations. Like *IXR1*, deletion of both copies of *RAD9* has a strong fitness benefit in diploids, but in this case there is a small fitness benefit in the *rad9Δ/RAD9* heterozygotes providing a path to inactivating the DNA damage checkpoint in which both steps increase fitness. This two-step path in diploids is less favorable than the one-step path that inactivates the checkpoint in haploids and likely accounts for the lower frequency of mutations affecting *RAD9* and other known checkpoint genes in evolved diploid clones (Fig 5 and S6 Data).

Two-step paths affected the replication machinery or DNA damage checkpoint in 9 of the 28 diploid clones; 2 involved independent mutations in the same gene and 8 produced homozygous mutations due to gene conversion or mitotic recombination (Fig 4, one clone contained two homozygous mutations). The high frequency of these events suggests that these mutations conferred some advantage even as heterozygotes. For one mutation, *sld5-E130K*, we verified the fitness benefit of the *sld5-E130K/SLD5* heterozygote, which suggests that the mutant subunit is present in a fraction of the genome-wide replicative helicases, where it helps to stabilize replication forks in the absence of Ctf4. Mitotic recombination can then homozygose the mutation, eliminating the wild-type Sld5 protein and stabilizing all replisomes. Ploidy thus explains both the reduced frequency of mutations in *RAD9* and dictates which mutations affecting DNA replication are ultimately selected (e.g. *IXR1* versus *SLD5*). We suggest that semi-dominant mutations are an important intermediate in the selection of loss-of-heterozygosity during the evolution of diploid, somatic cells.

## Molecular mechanisms affecting adaptation in recombination-deficient cells

Recombination-deficient clones have much fewer segmental amplifications affecting the genes for the cohesin loader, *SCC2* and *SCC4* (Fig 5C). The lack of the recombinase protein Rad52 in recombination-deficient strains dramatically reduces their ability to achieve segmental amplifications. Only one event, likely mediated by Rad52-independent recombination [105–108], was found in the recombination-deficient evolved clones (Fig 3B and S4 Table). The other mutations in these strains that are likely to affect sister chromatid cohesion were point mutations affecting other proteins involved in pairing and segregation of chromosomes such as Pds5 and Rad61 (Fig 5B). The higher number of point mutations in this and other modules is likely the consequence of increased spontaneous mutagenesis imposed by the lack of homology-mediated repair of spontaneously occurring DNA lesions (S5 Fig, [109]). Recombination proficiency thus explains both the reduced frequency of gene amplification affecting sister chromatid cohesion and the increased number of point mutations detected in evolved clones (Fig 6B).

## Impact of genomic features on evolutionary adaptation to constitutive DNA replication stress

Our analysis argues that although genomic features don't affect the ability of cells to adapt to DNA replication stress or the modules altered to produce adaptation, they do affect the mutational pathways by which strains with different genomic features adapt.

If the major adaptive mechanisms are shared, how do we explain the limited number of genes that are mutated in all conditions and the genes that are uniquely and significantly mutated in strains with specific genomic features (Fig 3A)? There are several hypotheses: some of these genes may be false positives, mutations in these genes confer benefits in more than one strain but were only detected in a single strain and mutations in these genes are substantially more, or exclusively, beneficial in one strain. The fact that our reconstructed strains recapitulate the fitness of the evolved populations with a minimal set of changes in genome maintenance suggests that feature-specific mutations play a minor role in adaptation. Of the 22 genes that were mutated in at least two strains, 10 have known roles in DNA replication, repair, or sister chromatid cohesion, whereas only 5 of the 28 that were uniquely mutated in one strain have such a role (p = 0.034, $\chi^2$ test). The fitness of the reconstructed strains (Fig 6A and 6B) also implies that the most important mutations for the adaptation to *ctf4Δ*-induced DNA replication stress are those that occur in modules that were mutated in all three strains (Fig 3A). Mutations that were never (*ixr1*), or rarely (*2xSCC2* and *rad9*) seen in a strain, could nevertheless increase its fitness. This finding suggests that caution should be used in inferring adaptive strategies from lists of genes that have been mutated more often than expected by chance, especially when comparing populations that were evolved under different conditions or have different ancestral genotypes. We argue that experimental dissection of adaptive trajectories is needed to distinguish the mutations that make major contributions to adaptation from minor players and mutational noise.

The distribution of mutations with known phenotypic effects suggests that genomic features control how much adaptation in different functional modules contributes to the overall response to selective pressure. Mutations that break the DNA damage checkpoint by inactivating *RAD9* are less frequent in diploid populations than they are in haploids, and amplifications of cohesin loader genes are less frequent in recombination-deficient populations than they are in recombination-proficient ones. Although we suspect that some of the diploid populations have found alternate genetic routes to partially or fully inactivate the DNA damage checkpoint and that some of the recombination-defective populations have selected other mutations that improve sister chromatid cohesion, these changes are likely to be less effective than those that were frequent in other combinations of genomic features.

## Implications for cancer and natural evolution

We previously speculated that evolutionary processes similar to the adaptation to constitutive DNA replication stress could assist the evolution of cancer cells and organisms in nature [17]. Tumor cells often experience DNA replication stress as a consequence of oncogene activation, promoting genetic instability and increasing the rate of cell death [19,22]. While generated through a different mechanism (unrestrained proliferation, rather than replisome perturbation), oncogene induced DNA replication stress produces cellular consequences [25] which are remarkably similar to the one described in the absence of Ctf4, such as the accumulation of ssDNA gaps, stalled and reversed forks [17,48,49], genetic instability [49,51,52] and DNA damage response activation [47,50]. Based on these similarities we speculate that the evolutionary adaptation to DNA replication stress could reduce its negative effects on cellular fitness and thus assist tumor evolution. In natural populations, interference with DNA replication either through the fixation of deleterious mutations, perturbation by selfish genetic elements or toxins secreted by antagonistic organisms could generate similar evolutionary adaptation. The consequences of these evolutionary processes could rewire genome maintenance modules to alter cancer cells' physiology and generate molecular diversity in nature [110]. For these scenarios to be plausible, the evolutionary adaptation to DNA replication stress needs to happen

over short timescales. In cancer, it would need to happen over the limited somatic evolution that accompanies cancer progression. In nature, it would need to occur before the perturbed organisms are outcompeted by unaffected ones or block the source of DNA replication stress. It could be argued that some genomic features, such as diploidy or reduced recombination, could reduce the likelihood of evolving new strategies for genome replication and maintenance. In our hands, however, these genomic features do not preclude rapid, evolutionary adaptation to constitutive DNA replication stress: loss of heterozygosity or increased mutation rates, which have both been documented in cancer [111,112] and natural evolution [113,114], allow cells to circumvent these obstacles and adapt quickly to a severe perturbation of the replication machinery.

The relevance of compensatory mutations for the subsequent evolution of cells depends on their effect when the initial genetic perturbation has been removed. We found that reintegrating a wild type *CTF4* gene into the evolved strains lead to fitness levels comparable to wild type cells in 8 of the 14 populations tested (S12A Fig). These results show that the combined effects of the mutations accumulated during the evolutionary adaptation have little effect once Ctf4 is restored, increasing the likelihood they would persist after reversal of the initial genetic perturbation. When the same populations were tested in the presence of Hydroxyurea, a drug which depletes nucleotide pools by inhibiting the ribonucleotide reductase, we found a diverse response, with populations to which Ctf4 had been restored being less (9/14), equally (1/14), and more (2/14) fit than wild type cells (S12B Fig). The adaptation to *ctf4Δ*-induced constitutive DNA replication stress can therefore be effective against other type of stresses perturbing DNA replication, but this is likely contingent on the particular mutations accumulated by the strains, and the nature of the replication defects applied to them. Interestingly, one rapidly evolving clade of yeasts has lost *RAD9* [115], the primary target of checkpoint-inactivating mutations in our experiments. Similarly, many cancers inactivate the DNA damage response during tumor evolution [116,117]. We suggest that the careful examination of the genomic sequences of tumors and fast-evolving species might reveal signs of adaptation that resemble the molecular evolution we have described.

## Materials and methods

### Strains

All strains were derivatives of a modified version (Rad5+) of S. cerevisiae strain W303 (*leu2-3,112 trp1-1 can1-100 ura3-1 ade2-1 his3-11,15, RAD5*). S11 Table lists each strain's genotype. The ancestors of wild-type, *ctf4Δ* and *ctf4Δ rad52Δ* strains were obtained by sporulating a *CTF4/ctf4Δ RAD52/rad52Δ* heterozygous diploid. This was done to minimize the selection acting on the ancestor strains before the beginning of the experiment. *CTF4/CTF4* and *ctf4Δ/ctf4Δ* diploid strains were obtained by mating freshly generated *MAT**a*** and *MAT*α haploid strains. The reintegration of a wild type *CTF4* gene in the genome of the evolved population was performed by transforming cells with an integrative plasmid (*YCplac211-URA3*) carrying the *CTF4* ORF and associated native regulatory regions. To represent the genetic diversity present within the populations, 50–100 colonies growing on each -URA plate were pooled in a unique sample (*CTF4*-recovered evolved population).

### Media and growth conditions

Standard rich medium, YPD (1% Yeast-Extract, 2% Peptone, 2% D-Glucose) was used for the evolution experiment and all fitness measurements.

## Experimental evolution

All populations used in the evolution experiment (EVO1-40) were initially inoculated in glass tubes containing 10 ml of YPD with yeast colonies of the relative genotype. Glass tubes were placed in roller drums at 30˚C and grown for 24 hr. Daily passages were done by diluting 10 μl of the previous culture into 10 ml of fresh YPD (1:1000 dilution, allowing for approximately 10 generations/cycle). All populations were passaged for a total of 100 cycles (≈1000 generations). In this regime, the effective population size is calculated as $N_0$ x $g$ where $N_0$ is the size of the population bottleneck at transfer and $g$ is the number of generations achieved during a batch growth cycle and corresponds to approximately to $10^7$ cells. After 1000 generations four evolved clones were isolated from the populations evolved in the absence of Ctf4 (EVO1-24) by streaking cells on a YPD plate (total of 96 clones isolated). To capture the within-population genetic variability we selected the clones displaying the largest divergence of phenotypes in terms of resistance to genotoxic agents (methyl-methanesulfonate, hydroxyurea and camptothecin). All final populations and clones isolated from them were mixed with 800 ml of 30% v/v glycerol and stored at -70˚C for future analysis. The evolution of haploid, diploid, and recombination defective strains, all lacking *CTF4*, and wild-type haploid and diploid strains were all carried out in parallel.

## Whole genome sequencing

Genomic DNA library preparation was performed with an Illumina (RRID:SCR_010233, San Diego, CA, US) Nextera DNA Library Prep Kit. Libraries were then pooled and sequenced either with an Illumina HiSeq 2500 (125bp paired end reads) or an Illumina NovaSeq (150 bp paired end reads). The SAMtools software package (RRID:SCR_002105, samtools.source-forge.net) was then used to sort and index the mapped reads into a BAM file. GATK (RRID: SCR_ 001876, www.broadinstitute.org/gatk; McKenna et al., 2010) was used to realign local indels, and VarScan (RRID:SCR_006849, varscan.sourceforge.net) was used to call variants. Clones and populations were sequenced at approximately the following depths: 25-30X for haploid clones, 50-60X for diploid clones, 50-60X for haploid populations and 120-130X for diploid populations. Mutations were found using a custom pipeline written in Python (RRID: SCR_008394, www.python.org). The pipeline (github.com/koschwanez/mutantanalysis) compares variants between the reference strain, the ancestor strain, and the evolved strains. Variants found in less than 25% and 35% of the reads in haploid and diploid populations respectively were discarded, since many of these corresponded to misalignment of repeated regions. For clone sequencing, only variants found in more than 75% of the reads in haploids and 35% of the reads in diploids (to account for heterozygosity) were considered mutations. The frequency of the reads associated with all the variants detected are reported in S1 Table. A variant that occurs between the ancestor and an evolved strain is labeled as a mutation if it either (1) causes a substitution in a coding sequence or (2) occurs in a regulatory region, defined as the 500 bp upstream and downstream of the coding sequence. Gapped alignments of the 150 paired-end reads in our data set permit the identification of small indels ranging in size from 1–55 bp using VarScan pileup2indel tool [118]. All small indels (and the respective sequence affected) are listed together with SNPs in S1 Table.

## Identification of putative adaptive mutations

To identify genes and cellular modules targeted by selection we used the following methods:

**Parallel evolution of genes.** This method relies on the assumption that those genes that have been mutated significantly more than expected by chance alone, represent cases of parallel evolution among independent lines. The mutations affecting those genes are therefore

considered putatively adaptive. The same procedure was used independently on the mutations found in wild-type and *ctf4Δ* evolved lines:

We first calculated per-base mutation rates as the total number of mutations in coding regions occurring in a given background, divided by the size of the coding yeast genome in bp (including 1000bp per ORF to account for regulatory regions)

$$\lambda = \frac{SNPs + indels}{bp\text{coding}}$$

If the mutations were distributed randomly in the genome at a rate λ, the probability of finding n mutations in a given gene of length N is given by the Poisson distribution:

$$\mathrm{P}\,(n\ mutations|gene\ of\ length\ N) = \frac{(\lambda N)^n e^{-\lambda N}}{n!}$$

For each gene of length N, we then calculated the probability of finding $\geq$ n mutations if these were occurring randomly.

$$P(\neq\geq n\ mutations|gene\ of\ length\ N) = \sum_{k=n}^{\infty} \frac{(\lambda N)^n e^{-\lambda N}}{k!} = 1 - \frac{\Gamma(n+1, \lambda N)}{n!}$$

(Where Γ is the upper incomplete gamma function) which gives us the p-value for the comparison of the observed mutations with the null, Poisson model. In order to decrease the number of false positives, we then performed multiple-comparison corrections. The more stringent Bonferroni correction (α = 0.05) was applied on the wild-type evolved mutations dataset, while Benjamini-Hochberg correction (α = 0.05) was used for the *ctf4Δ* mutation dataset. Genes that were found significantly selected in the evolved wild-type populations (after Bonferroni correction) were removed from the list of evolved *ctf4Δ* strains (genes mutated in wild-type evolved populations were removed from the *ctf4Δ* and *ctf4Δ rad52Δ* datasets and genes mutated in wild-type diploid populations were removed from the *ctf4Δ/ctf4Δ* dataset). We argue that since these genes were targets of selection even in wild-type cells, they are likely involved in processes that are unrelated to DNA replication and are instead associated with adaptation to sustained growth by serial dilutions. S2 Table lists the mutations detected in evolved clones and populations, after filtering out those that occurred in genes that were significantly mutated in the respective wild-type populations. Genes significantly selected are shown in dark grey (after Benjamini-Hochberg correction with α = 0.05).

**Parallel evolution of cellular modules.** Lists of genes that had been mutated more frequently than expected by chance (S2 Table) or found homozygously mutated in diploid strains (S6 Table) were further analyzed for gene ontology (GO) terms enrichment. Lists of mutations were input as 'multiple proteins' in the STRING database, which reports on the network of interactions between the input genes (https://string-db.org). The 'biological process' terms enriched in the significantly or homozygously mutated genes are listed in S3 and S7 Tables respectively.

## Gene ontology (GO) enrichment analysis

The list of genes with putatively selected mutations (Fig 3A) or homozygous mutations in diploids (Fig 4) were input as 'multiple proteins' in the STRING database, which reports on the network of interactions between the input genes (https://string-db.org). The GO term enrichment analysis provided by STRING are reported in S3 and S7 Tables respectively. Briefly, the strength of the enrichment is calculated as $Log_{10}(O/E)$, where O is the number of 'observed' genes in the provided list (of length N) which belong to the GO-term, and E is the number of

'expected' genes we would expect to find matching the GO-term providing a list of the same length N made of randomly picked genes. P-values are computed using a Hypergeometric test and corrected for multiple testing using the Benjamini-Hochberg procedure. The resulting P-values are represented as 'False discovery rate' in S3 and S7 Tables and describe the significance of the GO terms enrichment [119]."

### Fitness assays

To measure relative fitness, we competed the ancestors and evolved strains against reference strains. Cells were competed against reference strains with the same ploidy. Wild-type refence strains were used for the competition assays in all the figures, except for those in Fig 6A where *ctf4Δ* reference strains were used instead. A *pFA6a-prACT1-yCerulean-HphMX4* plasmid was digested with AgeI and integrated at one of the *ACT1* loci of the original heterozygous diploid (*CTF4/ctf4Δ RAD52/rad52Δ*) strain. This allows for the expression of fluorescent protein yCerulean under the strong actin promoter. The heterozygous diploid was then sporulated and dissected to obtain fluorescent wild-type, *ctf4Δ* or *ctf4Δ rad52Δ* reference haploid strains. *ctf4Δ/ctf4Δ* diploid reference strains were obtained by mating two *ctf4Δ* haploid strains, one of which expressed yCerulean under the *ACT1* promoter. For measuring a strain's relative fitness, 10 ml of YPD were inoculated in individual glass tubes with either the frozen reference or test strains. After 24 hrs the strains were mixed in fresh 10 ml YPD tubes at a ratio dependent on the expected fitness of the test strain compared to the reference (i.e. 1:1 if believed to be nearly equally fit) and allowed to proliferate at 30˚C for 24 hrs. 10 μl samples were taken from this mixed culture (day 0) and the ratio of the two starting strains was immediately measured. Tubes were then cultured following in the same conditions as the evolution experiment by diluting them 1:1000 into fresh medium every 24hrs for 2–4 days, monitoring the strain ratio at every passage. Samples were analyzed by flow cytometry (Fortessa, BD Bioscience, RRID: SCR_013311, Franklin Lakes, NJ, US). Strain ratios and the number of generations occurred between samples were calculated using a custom Python script utilizing the FlowCytometryTools package (https://pypi.org/project/FlowCytometryTools/). Ratios, *r*, were calculated based on the number of yCerulean positive and yCerulean negative events detected by the flow cytometer:

$$r = \frac{Non-yCerulean_{events}}{yCerulean_{events}}$$

Generations between time points, *g*, were calculated based on total events measured at time 0 hrs and time 24 hrs:

$$g = \frac{\log_{10}(Events_{t24}/events_{t0})}{\log_{10}2}$$

Linear regression was performed between the (*g*, $\log_e r$) points for each sample. Relative fitness was calculated as the slope of the resulting line. The mean relative fitness, *s*, was calculated from measurements obtained from at least three independent biological replicates. Error bars represent standard deviations. The P-values reported in figures are the result of t-tests assuming unequal variances (Welch's test).

### Copy number variations (CNVs) detection by sequencing

Whole genome sequencing and read mapping was done as previously described. The read-depths for every unique 100 bp region in the genome were then obtained by using the VarScan

copynumber tool. A custom pipeline written in python was used to visualize the genome-wide CNVs. First, the read-depths of individual 100 bp windows were normalized to the genome-wide median read-depth to control for differences in sequencing depths between samples. The coverage of the ancestor strains was then subtracted from the one of the evolved lines to reduce the noise in read depth visualization due to the repeated sequences across the genome. The resulting CNVs were smoothed across five 100 bp windows for a simpler visualization. Final CNVs were then plotted relative to their genomic coordinate at the center of the smoothed window. Since the wild-type CNVs were subtracted from the evolved CNVs, the y axis refers to the copy number change occurred during evolution (i.e. +1 means that one an extra copy of a chromosome fragment has been gained in haploid cells, and that two extra copies have been gained in diploids). Segmental amplifications were identified as continuous chromosomal regions affected by the same change in copy number (Fig 3B and S4 Table). Amplified genes (Fig 5A and 5C) were obtained from the lists of ORFs included within the boundaries of the segmental amplifications.

## The correlation of chromosomal features with segmental amplification boundaries

We first identified the approximate boundaries (± 500bp) of all segmental amplification by manually inspecting the BAM files with the Integrative Genomic Viewer (https://software. broadinstitute.org/software/igv/). The coordinates of the sequence features that define the boundary between lower and higher read-depths are reported in S4 Table. We then examined the sequences within these approximate boundaries to determine whether various chromosomal features were over- or underrepresented. We considered features that previous studies have found to be associated with hotspots for lesions and sources of genetic instability. We first counted how many times a given feature fell in the boundary zones of segmental amplifications. Then we calculated the expected number of features in these zones based on the total number of features in the genome and the percentage of the genome represented by segmental amplification boundaries zones. We compared these numbers by $\chi 2$ analysis and reported the associated p-values (S8 Table). The number of tRNA genes, transposable elements, LTRs, ARS elements, snRNA and snoRNA genes and centromeres in the genome were determined using YeastMine (https://yeastmine.yeastgenome.org/). Coordinates of fragile sites were obtained from [120]. G4 sequences were obtained from [121]. Highly-(top 5%) and weakly-(least 5%) transcribed genes were identified from the data in [122]. Rrm3 binding sites and regions with high levels of γH2AX were derived from [123] and [124], respectively. Sites of DNA replication termination were derived from [17]. The tandemly repeated sequences, with a minimal repeat tract of twenty-four bases, were obtained from the tandem-repeat-database (TRDB; https:// tandem.bu.edu/cgi-bin/trdb/trdb.exe).

## Supporting information

**S1 Fig. Fitness of the evolved clones.** Fitness of 96 clones isolated from the 24 evolved populations (4 clones for population, distributed on the same vertical line), relative to haploid or diploid WT cells ($s = 0$). Fitness data of haploid strains (orange) is from [17]. Semi-transparent grey dots represent clones that became haploid over the course of the experiment. Error bars represent standard deviations. **a** and α refer to the strains' mating type (*MAT* locus). **a**/α indicates diploid strains. The P-values reported in figures are the result of t-tests assuming unequal variances (Welch's test). The fitness values shown here are reported in S2 Data.
(TIF)

**S2 Fig. Fitness of the populations evolved in the absence of DNA replication stress.** Fitness of the WT haploids (white) and diploids (black) ancestors and of 16 evolved populations derived from them (four from each ancestor), relative to WT cells ($s = 0$). Fitness data of haploid strains (white) is from [17]. Semi-transparent gray dots represent populations that changed ploidy over the course of the experiment. Error bars represent standard deviations. **a** and α refer to the strains' mating type (*MAT* locus). **a**/α indicates diploid strains. The P-values reported in figures are the result of t-tests assuming unequal variances (Welch's test). The fitness values shown here are reported in S1 Data.
(TIF)

**S3 Fig. Fitness increase of the evolved populations relative to the ploidy of the ancestor cells.** Fitness increase over 1000 generations (Δ*s*, measured as the difference between the populations' final fitness and the fitness of their respective ancestors that lacked Ctf4) relative to the ploidy of the ancestor cells (N). Fitness data of haploid strains (orange and white) is from [17]. Error bars represent standard deviations. The P-values reported in figures are the result of t-tests assuming unequal variances (Welch's test). The data shown here are reported or derived from S1 Data.
(TIF)

**S4 Fig. Fraction of the initial fitness defect recovered during the experiment. (A)** The populations' fitness increase over 1000 generations (Δ*s*) was divided by their ancestors' fitness defect (relative to strains of the same ploidy containing Ctf4) to calculate the fraction of the initial defect that was recovered during the experiment. **(B)** Fraction of the initial fitness defect recovered over the course of the experiment, relative to the ancestor's fitness. Fitness data of haploid strains (orange) is from [17]. Error bars represent standard deviations. The P-values reported in figures are the result of t-tests assuming unequal variances (Welch's test). The data shown here are reported in or derived from S1 Data.
(TIF)

**S5 Fig. Number and type of mutations found in the evolved clones. (A)** Total number of mutations per haploid genome detected in the clones of each of the three strains with different genomic features. **(B)** Total number of synonymous mutations per haploid genome for each strain. **(C)** Total number of segmental amplifications per haploid genome for each strain. The P-values reported in figures are the result of t-tests assuming unequal variances (Welch's test). The values shown in A and B are reported in S4 Data. Values shown in C are derived from S4 Table.
(TIF)

**S6 Fig. Putative adaptive mutations in evolved wild type populations. (A)** Venn diagram representing the mutations found in the wild-type evolved populations. The circles' areas are proportional to the unique list of genes found mutated in the haploid or diploid evolved WT populations. 14 genes were mutated in both diploids and haploids and 13 were mutated only in haploids. **(B)** Venn diagram of the genes in which mutations were significantly selected. Notes that all the 14 mutations detected in the evolved diploids were also present in the evolving haploids populations (panel A), but the 9 genes listed in most right sector in panel B were only significantly selected in evolved diploids.
(TIF)

**S7 Fig. Amplification of cohesin loader genes in evolved populations. (A)** Venn diagram representing the number of clones carrying *SCC2* (yellow) and *SCC4* (green) amplifications. Note how all clones carrying an *SCC4* amplification also have a *SCC2* amplification, and that

out of the 25 clones that had amplified both genes, 24 were detected in diploid populations.
**(B)** Cartoon of our model for the amplification of cohesion loaders during adaptation to constitutive DNA replication stress: Scc2 is initially the limiting subunit for the formation of active complexes. The initial amplification of *SCC2* produces extra Scc2 protein and thus increases the number of active cohesin loaders and makes Scc4 the limiting component. The subsequent amplification of *SCC4* provides extra binding partners and further increases the number of active complexes available.
(TIF)

**S8 Fig. Frequencies of mutations detected in the evolved clones.** Frequencies of mutations in **(A)** *ctf4Δ* **(B)** *ctf4Δ/ctf4Δ* and **(C)** *ctf4Δ rad52Δ* evolved clones were obtained from the ratios of mutated to total DNA reads covering the locus. Mutations present in more than 77% of the reads were considered homozygous (Hom) in *ctf4Δ/ctf4Δ* diploids. Mutations present in less than 77% of the reads were considered heterozygous (Het). **(D)** Number of heterozygous (light blue) and homozygous mutations (dark blue) found in each individual, sequenced, evolved *ctf4Δ/ctf4Δ* diploid clone. All the values shown here are derived from S1 Table.
(TIF)

**S9 Fig. Estimated frequency in the evolved populations of clones carrying homozygous mutations.** Clonal frequency estimates within populations were derived from the percentage of reads from whole population sequencing that carried those mutations previously identified, in clones, as homozygous. All the values shown here are derived from S1 Table.
(TIF)

**S10 Fig. Segmental amplification boundaries correspond to repetitive sequences.** Magnification on the most recurrent copy number variations (CNVs) affecting different chromosomes (roman numbers). Repetitive sequences present in the surrounding chromosomal coordinates are noted in different colors: tRNA genes in yellow, Long Terminal Repeats (LTRs) in green, transposons in purple and *MAT* loci in red. Copy number change refers to the fragment's gain or loss during the evolution experiment (i.e. +1 means that one copy was gained in haploid cells, and that two copies were gained in diploid cells).
(TIF)

**S11 Fig. Limited evidence for diminishing return epistasis.** The fitness increase provided by four adaptive mutations reconstructed in strains with different genomic features versus their genetic background's ancestral fitness defect. Fitness data of haploid strains (orange) is from [17]. Error bars represent standard deviations. The fitness values shown here are reported in S6 Data.
(TIF)

**S12 Fig. Effect of compensatory mutations after restoring Ctf4.** A wild type *CTF4* gene was reintegrated into the genome of evolved cells. The integration was not possible in populations 2,3 and 11, which became prototrophic for the marker used for the integration (*URA3*) and in all recombination-deficient strains, because of their inability to recombine an exogenous DNA fragment into their genome. **(A)** Fitness of the *CTF4*-restored evolved populations in YPD (left panel). Fitness of the *CTF4*-restored and empty-vector transformed ancestors (right panel). **(B)** Fitness of the *CTF4*-recovered evolved populations in the presence and absence of 50mM Hydroxyurea. The points in the absence of hydroxyurea also appear in A. The fitness values shown here are reported in S9 Data.
(TIF)

**S1 Table. Mutations detected in evolved populations and clones.**
(XLSX)

**S2 Table. Putative adaptive mutations.**
(XLSX)

**S3 Table. GO term enrichment analysis of putative adaptive mutations.**
(XLSX)

**S4 Table. Segmental amplifications detected in evolved clones.**
(XLSX)

**S5 Table. Enrichment analysis of genes present on segmental amplifications.**
(XLSX)

**S6 Table. Homozygous mutations detected in evolved diploids.**
(XLSX)

**S7 Table. GO term enrichment analysis of homozygous mutations.**
(XLSX)

**S8 Table. Chromosomal features enriched in proximity of segmental amplification boundaries.**
(XLSX)

**S9 Table. Mutations affecting *IXR1*.**
(XLSX)

**S10 Table. Mutations affecting *RAD9*.**
(XLSX)

**S11 Table. List of strains employed in the study.**
(XLSX)

**S1 Data. Source data for Fig 1.**
(XLSX)

**S2 Data. Source data for S1 Fig.**
(XLSX)

**S3 Data. Source data for Fig 2A.**
(XLSX)

**S4 Data. Source data for S5 Fig.**
(XLSX)

**S5 Data. Source data for Fig 3.**
(XLSX)

**S6 Data. Source data for Fig 5.**
(XLSX)

**S7 Data. Source data for Fig 6A.**
(XLSX)

**S8 Data. Source data for Fig 6B.**
(XLSX)

**S9 Data. Source data for S12 Fig.**
(XLSX)

## Acknowledgments

We thank Michael Laub, María Angélica Bravo Núñez, Sriram Srikant, Thomas LaBar, Andrea Giometto and Mariana Natalino for critical reading of the manuscript; Andrea Giometto for assistance in data analysis; Veronika Fitz, Claire Hartman from the Harvard Bauer Core Facility, Beatriz Teixeira and Denise Brito from the IGC flow cytometry facility for technical assistance; We thank the members of the Murray and Nelson labs for helpful discussions.

## Author Contributions

**Conceptualization:** Marco Fumasoni, Andrew W. Murray.

**Data curation:** Marco Fumasoni.

**Formal analysis:** Marco Fumasoni.

**Funding acquisition:** Marco Fumasoni, Andrew W. Murray.

**Investigation:** Marco Fumasoni.

**Project administration:** Marco Fumasoni, Andrew W. Murray.

**Supervision:** Andrew W. Murray.

**Writing – original draft:** Marco Fumasoni.

**Writing – review & editing:** Marco Fumasoni, Andrew W. Murray.

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
