## [Decision Letter · Decision Letter 0]

30 Apr 2021

Dear Dr Fumasoni,

Thank you very much for submitting your Research Article entitled 'Genome architecture shapes evolutionary adaptation to DNA replication stress' to PLOS Genetics.

The manuscript was fully evaluated at the editorial level and by two independent peer reviewers. The reviewers appreciated the attention to an important problem, but raised some substantial concerns about the current manuscript. Specifically, reviewer 2 felt that the advance represented by the current manuscript is minor and suggested several possible avenues for improvement. Based on the reviews, we will not be able to accept this version of the manuscript, but we would be willing to review a much-revised version. We cannot, of course, promise publication at that time.

If you decide to revise the manuscript for further consideration at PLOS Genetics, please aim to resubmit within the next 60 days, unless it will take extra time to address the concerns of the reviewers, in which case we would appreciate an expected resubmission date by email to plosgenetics@plos.org.

[LINK]

We are sorry that we cannot be more positive about your manuscript at this stage. Please do not hesitate to contact us if you have any concerns or questions.

Yours sincerely,

Jianzhi Zhang

Associate Editor

PLOS Genetics

Kirsten Bomblies

Section Editor: Evolution

PLOS Genetics

Reviewer's Responses to Questions

**Comments to the Authors:**

Reviewer #1: This work is a follow-up of the authors’ previous work. Previously, the authors studied how haploid yeast cells adapted to DNA replication stress (CTF4 deletion) using experimental evolution. In the current work, the authors did the same experiment in two other genetic backgrounds (diploid cells and recombination-deficient haploid cells). Overall, the adaptive mutations accumulated in the three strains share some similarity, this is further proven by Figure 6, where the fitness of the ancestors is fully restored by introducing the same set of mutations to all three strains. However, the authors also found some adaptive mechanisms that are specific to each genetic background. For example, the authors reported the “two-step mutations” (a mutation followed by a LOH event) in diploids and there is almost no gene amplification in recombination-deficient strain. In conclusion, I think this is a solid work that deserves a publication and I recommend a minor revision. My specific questions are listed below:

1. The authors define ploidy and rad52-deletion as “genome architecture”. I think they are just different genetic backgrounds, not different genome architectures in the strict sense. I agree with the definition of genome architecture I found online: “Genome architecture describes the spatial arrangement of the functional and regulatory elements of a genome.” I’m not an expert on terminology, but I think it would be better to use a more accurate term.

2. In lines 133-135, the authors mentioned that wildtype haploid and diploid are evolved in the same environment to control for mutations that confer environment-specific benefit. But this data is not used in the later analysis.

3. Does CTF4 deletion affect spontaneous mutation rate? Maybe compare the number of mutations accumulated in CTF4-deletion strain with the wildtype.

4. The mutation rate of the rad52 deletion strain is much higher than the other two, this feature should also contribute to the faster evolution of the recombination-deficient strain. This point should be mentioned.

5. The citation of Figure 2C (line 171) in the MS appears early than Figure 2B (line 178)

Reviewer #2: In the current manuscript, Fumasoni and Murray investigate how baker’s yeast can adapt to constitutive DNA replication stress, a physiological condition that is frequently observed in cancer cells. More specifically, they ask how the ploidy or the recombination competence of the yeast affect its adaptive potential and the mechanisms underlying adaptation to DNA replication stress. In order to trigger constitutive DNA replication stress, they deleted a gene (CTF4) that encodes a non-essential subunit of the DNA replication machinery. As expected, this genetic perturbation causes a significant decrease in reproductive fitness, providing an opportunity to explore genomic mutations that accumulate during a microbial evolution experiment and compensate the fitness loss. The Authors evolved two different types of strains: i.) a diploid yeast and ii.) a haploid yeast deficient in homologous recombination (due to the lack of RAD52). By characterizing the evolved strains of this current study, the Authors were able to compare the evolutionary trajectories of the above strains to that of those recombination-proficient haploids that were previously adapted by the same Authors using the very same pipeline (PMID: 32043971).

The main results are the followings:

1. At the end of the evolutionary experiment (after 1000 generations), all parallel evolved lines of all genotypes showed fitness compensation (i.e. displayed fitness increase) in the presence of DNA replication stress.

2. The extent of fitness increase showed a negative correlation with the initial fitness of the original strains, providing further evidence for the declining adaptability concept (e.g. PMID: 25815007).

3. To shed light on the mechanisms of adaptation towards DNA replication stress, whole genome sequencing was performed on the evolved lines. The Authors found several mutations, a larger fraction of them being unique to the type of strains, whereas a smaller fraction of them showed convergence at the functional modules across the strains.

4. Specifically, the evolved lines of all three types of strains (two in the current, one in the previous study) harbored genomic alterations in three biological processes, including DNA replication, DNA damage checkpoint and sister chromatid cohesion. Hence, the principles that govern adaptation to DNA replication stress are similar across all tested strains, irrespective of ploidy and recombination competence. Nevertheless, the exact genes that were mutated, and the frequency of different types of mutations largely depended on the genetic background. For instance, amplification events were less frequent in the recombination-deficient strains and recessive null mutations were rarer in diploids.

5. One-by-one reconstruction of several single mutations, affecting either of the abovementioned functional modules, increased the fitness of the ancestor strains in all cases, albeit different mutations conferred different benefits. However, the fitness benefit these mutations provided was similar across all types of strains, at least in the majority of tested mutations (3 out of 4). In addition, combination of these mutations in the ancestor strains resulted in a fitness that is highly similar to that of the evolved strains, indicating that a limited number of mutations in genome maintenance was enough to fully mirror the evolved fitness.

To sum up, the current study line-up an experimental evolution in the lab to investigate adaptation of baker’s yeast towards DNA replication stress. Nevertheless, this is something that was already done in their previous paper (PMID: 32043971), although in a different genetic background. I do understand that the main motivation of the Authors was to compare the adaptation mechanisms of genotypes with different genomic characteristics (in terms of ploidy/recombination competence), but the results are somewhat expected: different strains preferentially accumulated different types of genomic mutations, at a different frequency. As the initial stress was the same in all backgrounds, mutations in genes belonging to the same genome maintenance modules is also something that is not so surprising. That being said, I have ambivalent feelings about the novelty of the current study from a conceptual point of view: it feels like that this work is just a follow-up study of a previous idea (PMID: 32043971, with two additional types of strains being tested) with no major conceptual advance. In addition, I also feel that the Authors have not put enough efforts into the current study that would be comparable to what they did in their previous work (PMID: 32043971).

For instance, the present study do not line up any detailed biological assays to measure the effect of compensatory mutations on chromatid separation/DNA replication, etc.; something that was present in their previous work (PMID: 32043971). In addition I failed to find a thorough examination of the potential side-effects of the compensatory mutations in other genomic backgrounds/conditions (see major comment 4 below) that would be essential to assess their long-term prevalence in natural environmental settings.

Major points/questions:

1. The term genome architecture is traditionally used for the arrangement of genes, regulatory elements in the genome (PMID: 18929678) or to refer to the spatial arrangement of chromosomes within the nuclear volume (https://doi.org/10.1016/S1067-5701(98)80021-1). However, in this work (line 44-46), it is used to refer to the “ploidy, rates of recombination, and the activity of mobile genetic elements”, which is a bit odd to this Reviewer. Since this term is also used in the title, it can be misleading for the scientific community, as spatial arrangement of genes/chromosome was not tested in this work at all. I suggest replacing this term with a more appropriate one. On a side note, the Authors also mention the activity of mobile genetic elements among the features of genome architecture, but obviously, adaptation of a strain that has a modified activity of mobile genetic elements has not been performed here.

2. The extent of compensation differed across the tested strains: recombination-deficient haploid strains showed the largest fitness increase, followed by diploids and recombination-competent haploid strains. This difference can be accounted for the declining adaptability (less fit genotypes are more likely to get compensated for than those that are closer to the WT fitness). This is all OK. However, in my opinion, the Authors might misinterpret this observation. They claim that diploids evolve faster than haploids (faster appears multiple times throughout the manuscript), but we have no firm evidence for that. The extent of compensation is not the same as the speed of adaptation. I might be wrong, but to make inferences about the speed of adaptation, one would need to investigate fitness at multiple time points during the evolutionary experiment and compare the kinetics of the fitness increase across the tested strains. By doing that, one would also be able to see if there is any saturation in the fitness increase during the 1000 generations, or it is still inclining. Without doing that, we cannot objectively compare the speed of adaptation across different types of strains.

3. The Authors investigated the effect of segmental amplifications by focusing on two genes that reside on the observed segmental amplifications, namely SCC2 and SCC4, members of the cohesin loader complex. By introducing an additional copy of SCC2 into the ancestor strains, the Authors confirmed the compensatory role of this amplification, although the fitness benefit it provides seems to be a minor one compared to that of other mutations. This finding is actually rather surprising as such an increase in the gene copy number of a single subunit might have caused a stoichiometric imbalance in the complex and a subsequent fitness loss, but apparently this is not the case here. I might be wrong, but one might expect that amplifications involving both subunits of the SCC2-SCC4 complex should make the complex more functional compared to a situation when only one is amplified. Nevertheless, it would be useful to report the number of cases (e.g. not the total, but the mean/median of all lines per backgrounds), where only a single gene or both genes were amplified in a given background during the lab evolution.

4. Relating to the segmental amplifications, I have also another issue that is the involvement of several linked genes residing on the same genomic segment. Which genes were involved in these amplifications (apart from SCC genes)? Do we have any functional enrichment on these segments? How stable are these genomic alterations? Are there dosage sensitive genes on these segments? What could be the potential side-effects arising from the increased copy number of these linked genes? Do they affect fitness, or any other phenotypic traits, in the lack of the original DNA replication stress? Actually, all of these questions about the fitness cost of mutations can be applied to other genomic alterations, including SNPs as well. These questions could be readily investigated by introducing these mutations into the WT strain one-by-one, or alternatively, by reintroducing the deleted CTF4 gene into the evolved lines (and hence investigating the sum effect of all compensatory mutations). On a related note, I believe it would be also important to demonstrate that the beneficial effects of the compensatory mutations are not contingent on the deletion of CTF4, but to DNA replication stress in general. In other words, the Authors should test the effect of compensatory mutations in the presence of DNA replication stress triggered by other genetic (deleting other important player of DNA replication) or conditional means (using drugs that induce the stress).

5. The Author’s main focus is on the recurrently mutated genes that accumulated in all three types of strains. They also demonstrated that these mutations, representing only a small fraction of all mutations, are largely responsible for the fitness compensation. For this reason, they argue that the large fraction of the mutations, those that are strains-specific, play only a marginal role in adaptation. On a side note, when they reconstructed such mutations (affecting one of the three genome maintenance modules) that were never seen in a given background, they happened to see beneficial fitness effect. Thus, I would not exclude the possibility that emergence of unique mutations only on one genomic background (including those that are not involved in the genome maintenance) would not be relevant from an evolutionary point of view. In my opinion, it would be informative to expand the text accordingly by adding a short overview of the type of genes that were uniquely mutated in each specific type of strains. This is something that is much expected when the Authors would like to pursue the differences of adaptation between different genetic backgrounds.

Minor points/questions

1. Is there any overlap of the mutated genes and those that are in suppressor interaction with CTF4?

2. The Authors mention that diploidization is frequently observed during experimental evolution. Have the Authors investigated this possibility in the case of their haploid strains (both in the current and the previous study)?

3. In order to measure fitness, The Authors used competition experiment. For this purpose, they introduced a fluorescent reporter into the promoter region of the ACT1 gene (a gene being both haploinsufficient and essential) of the strains. Have the Authors tested if insertion into this genomic locus does not perturb actin function and does not have any side-effects?

**Have all data underlying the figures and results presented in the manuscript been provided?**

Reviewer #1: Yes

Reviewer #2: Yes

PLOS authors have the option to publish the peer review history of their article (what does this mean?). If published, this will include your full peer review and any attached files.

Reviewer #1: No

Reviewer #2: No

---

## [Decision Letter · Decision Letter 1]

13 Oct 2021

Dear Dr Fumasoni,

We are pleased to inform you that your manuscript entitled "Ploidy and recombination proficiency shape the evolutionary adaptation to constitutive DNA replication stress" has been editorially accepted for publication in PLOS Genetics. Congratulations!

Please the note the minor comment by reviewer 1 and address it as you see fit in the final file for publication.

Yours sincerely,

Jianzhi Zhang

Associate Editor

PLOS Genetics

Kirsten Bomblies

Section Editor: Evolution

PLOS Genetics

Comments from the reviewers (if applicable):

Reviewer's Responses to Questions

**Comments to the Authors:**

Reviewer #1: I appreciate the effort that the authors spent in this revision. I have only one question, which is a follow-up on my previous 4th question.

I don’t agree with the two points made here by the authors. First, the fitness of ctf4-rad52-double-deletion strain apparently saturated after 100 generations. So, comparing the difference in fitness in 100-1000 generations seems not fair. Second, the mutation rate in diploids is not higher per se, but they do accumulate more mutations than haploid.

I think it’s probably true that 1) strains with lower initial fitness evolve faster, and 2) strains with higher mutation rate also evolve faster. But in the case of ctf4-rad52-double-deletion strain, it is not clear which is the primary cause. I would recommend the author to modify the content here (lines 481-489).

There is also a key typo in the reply but not in the main text.

“Second, slower (it is “faster” in the text) adaptation in diploids……..”

**Have all data underlying the figures and results presented in the manuscript been provided?**

Reviewer #1: None

PLOS authors have the option to publish the peer review history of their article (what does this mean?). If published, this will include your full peer review and any attached files.

Reviewer #1: No

**Data Deposition**

http://datadryad.org/submit?journalID=pgenetics&manu=PGENETICS-D-21-00445R1

**Press Queries**

---

## [Editor Report · Acceptance letter]

3 Nov 2021

PGENETICS-D-21-00445R1 

Ploidy and recombination proficiency shape the evolutionary adaptation to constitutive DNA replication stress 

Dear Dr Fumasoni, 

We are pleased to inform you that your manuscript entitled "Ploidy and recombination proficiency shape the evolutionary adaptation to constitutive DNA replication stress" has been formally accepted for publication in PLOS Genetics! Your manuscript is now with our production department and you will be notified of the publication date in due course.

With kind regards,

Andrea Szabo

PLOS Genetics

On behalf of:
